# Laser-Induced Morphological and Structural Changes of Cesium Lead Bromide Nanocrystals

**DOI:** 10.3390/nano12040703

**Published:** 2022-02-20

**Authors:** Athanasia Kostopoulou, Konstantinos Brintakis, Maria Sygletou, Kyriaki Savva, Nikolaos Livakas, Michaila Akathi Pantelaiou, Zhiya Dang, Alexandros Lappas, Liberato Manna, Emmanuel Stratakis

**Affiliations:** 1Institute of Electronic Structure and Laser, Foundation for Research and Technology-Hellas, 71110 Heraklion, Greece; masyg@iesl.forth.gr (M.S.); savvak@iesl.forth.gr (K.S.); nlivakas@gmail.com (N.L.); akathi39@gmail.com (M.A.P.); lappas@iesl.forth.gr (A.L.); 2Nanochemistry, Istituto Italiano di Tecnologia, 16163 Genova, Italy; dangzhy3@mail.sysu.edu.cn (Z.D.); liberato.manna@iit.it (L.M.); 3Department of Physics, University of Crete, 71003 Heraklion, Greece

**Keywords:** perovskite nanocrystals, photo-induced structural change, femtosecond laser, anion exchange

## Abstract

Metal halide perovskite nanocrystals, an emerging class of materials for advanced photonic and optoelectronic applications, are mainly fabricated with colloidal chemistry routes. On the quest for new properties according to application needs, new perovskite systems of various morphologies and levels of doping and alloying have been developed, often also involving post-synthesis reactions. Recently, laser irradiation in liquids has been utilized as a fast method to synthesize or transform materials and interesting laser-induced transformations on nanocrystals were induced. These studies in general have been limited to small nanocrystals (~15 nm). In the case of halide perovskites, fragmentation or anion exchange have been observed in such laser-based processes, but no crystal structure transformations were actually observed or deliberately studied. Nanocrystals are more sensitive to light exposure compared to the corresponding bulk crystals. Additional factors, such as size, morphology, the presence of impurities, and others, can intricately affect the photon absorption and heat dissipation in nanocrystal suspensions during laser irradiation. All these factors can play an important role in the final morphologies and in the time required for these transformations to unfold. In the present work, we have employed a 513 nm femtosecond (fs) laser to induce different transformations in large nanocrystals, in which two phases coexist in the same particle (Cs_4_PbBr_6_/CsPbBr_3_ nanohexagons of ~100 nm), dispersed in dichlorobenzene. These transformations include: (i) the exfoliation of the primary nanohexagons and partial anion exchange; (ii) fragmentation in smaller nanocubes and partial anion exchange; (iii) side-by-side-oriented attachment, fusion, and formation of nanoplatelets and complete anion exchange; (iv) side-by-side attachment, fusion, and formation of nanosheets. Partial or complete Br-Cl anion exchange in the above transformations was triggered by the partial degradation of dichlorobenzene. In addition to the detailed analysis of the various nanocrystal morphologies observed in the various transformations, the structure–photoluminescence relationships for the different samples were analyzed and discussed.

## 1. Introduction

Photon-induced processes, including photothermal, photochemical or photophysical transformations have been carried out in colloids in order to prepare nanocrystals of different morphologies or to modify the size or morphological features of pre-synthesized nanocrystals [1]. Laser irradiation, in particular, has been exploited to modify the nanocrystal size of various materials through a fragmentation process [2,3,4], or to alter their shape [5,6,7,8,9]. Furthermore, nanocrystals of different chemical phases have been obtained by ablation-based processes from bulk-like materials of similar stoichiometry [10,11,12] or by photon-triggered reactions in the corresponding precursors/reactants solutions [13,14]. In addition, laser-induced melting (mainly used to prepare bimetallic alloys) [15,16,17] or doping [18,19] in liquids have been proposed as effective methods to modify the nanocrystal stoichiometry. Finally, laser-triggered processes have been introduced to couple together two different nanomaterials, with examples including graphene, carbon nitride sheets and carbon nanotubes conjugated with different types of metallic and semiconducting nanocrystals [20,21,22,23].

In the specific case of metal halide perovskite nanocrystals, the research has been focused on understanding the origin of their interesting size- and shape-dependent optoelectronic properties, which are responsible for their unique applications in energy conversion and storage, sensing, and light emission [24,25,26,27,28]. So far, only a few works have been reported on laser-based fabrication or size/structure modification of metal halide perovskite nanocrystals, hence we still have a limited knowledge of photon–matter interactions in such materials. Published works have concerned the pulsed laser fragmentation of small nanocubes in a liquid environment [3,29] and the alteration of their stoichiometry via anion exchange with the halides originated from the solvent [30]. Shape modifications from a platelet-like morphology to bulk-like structures deposited on substrates have been also observed by tuning the excitation conditions, such as continuous wave or pulsed light irradiation, photon energy, pulse duration, and repetition rate [31]. Notably, a reversible, photo-induced orthorhombic-to-cubic phase transition has been recently demonstrated [32]. Despite the progress in this direction, further research is required to better understand the interaction of the laser photons with perovskite nanocrystals of more complex structures and morphologies in liquid dispersions.

In this report, the photon–nanocrystal interactions have been studied using Cs_4_PbBr_6_/CsPbBr_3_ nanocrystals of around 100 nm in size dispersed in a chlorinated solvent (dichlorobenzene). We have studied the dependence of the morphology and crystal structure of the final nanocrystals on the laser fluence and wavelength. Various laser fluences have been implemented in order to optimize the parameters that could lead to well-defined morphologies and good crystallinity after the irradiation. The study of the photo-induced structural or morphological transformations of lead halide perovskites (Figure 1) is important from both the fundamental and practical points of view. Specifically, this work contributes to the study of the impact on structure and morphology and thus to the PL properties of cesium lead halide nanocrystals. The work also provides a room-temperature rapid method to modify or tune the nanocrystal features. In the timeframe of minutes, various nanocrystal morphologies can be obtained, without the need of any further treatment or purification. Figure 1 summarizes the different transformations that took place over irradiation time in the liquid dispersion of the nanohexagons by employing a 513 nm fs laser.

## 2. Materials and Methods

### 2.1. Preparation of the Nanocrystal Colloidal Solution for the Irradiation

In this work, colloidal dispersions of hexagonally shaped Cs_4_PbBr_6_/CsPbBr_3_ nanocrystals, synthesized via a low-temperature reprecipitation-based protocol previously reported by our group, have been used as starting materials [33]. Following the synthesis, the pre-formed lead halide nanocrystals were separated upon centrifugation at 1000 rpm for 5 min and finally redispersed in 1,2-dichlorobenzene (DCB, spectrophotometric grade, 99%, Sigma Aldrich, Saint Luis, MI, USA) and used for the irradiation experiments. 

### 2.2. Laser Irradiation Experiment: Set Up and Irradiation Conditions

Details on the laser setup used for the photo-induced alterations of the Cs_4_PbBr_6_/CsPbBr_3_ nanocrystals can be found in our previous work [21]. All the irradiation experiments were carried out in ambient conditions at room temperature by placing the DCB dispersion of nanocrystals in a quartz cuvette. A high repetition rate femtosecond laser system using a directly diode-pumped Yb:KGW (ytterbium doped potassium gadolinium tungstate) as active medium was employed to irradiate the solutions. The laser source emitted linearly polarized pulses of 170 femtoseconds at 60 kHz repetition rate, at 513 nm wavelength. The laser beam was guided, through proper optical mirrors and an optical lens with focal length of 20 cm, to the sample that was placed 3 mm out of the focal point. The laser fluence was adjusted from 0.5 to 129 mJ/cm^2^, while the number of the pulses ranged from a single pulse to 57.6 × 10^6^ pulses. It is important to notice that 1.8 × 10^6^ pulses correspond to 0.5 min of irradiation, while 57.6 × 10^6^ pulses to 16 min. The irradiation experiments were repeated five times to check the reproducibility with samples from the same nanocrystal batch or from a different one.

In order to optimize the parameters that could lead to well-defined morphologies and good crystallinity after the irradiation, different fluences have been tested before choosing the value of the 129 mJ/cm^2^. Irradiation with a smaller (92 mJ/cm^2^) and a larger fluence (165 mJ/cm^2^) have been tested and the best conditions were decided according to the quality of the crystals from TEM images and their PL properties. IR wavelengths were also used, while keeping all the other irradiation conditions the same. These laser wavelengths however are not efficient to induce obvious morphological nanocrystal transformations, possibly due to the poor absorption of nanocrystals at such energies. The discussion of these data is included in the Appendix A. 

### 2.3. Characterization of the Nanocrystals

The morphological and structural features of the primary and irradiated nanocrystals have been studied using a LaB6 JEOL 2100 high resolution transmission electron microscope (JEOL Ltd., Akishima, Tokyo, Japan) operating at an accelerating voltage of 200 kV. Low magnification and HRTEM images of the nanocrystals were recorded on a Gatan ORIUS TM SC 1000 CCD camera (Gatan Inc., Pleasanton, CA, USA). For this experiment, the sample preparation includes a drop casting of the pristine and the irradiated solution onto a carbon-coated copper TEM grid, followed by solvent evaporation. The structural features of the nanocrystals were studied through FFT patterns obtained from HRTEM images. 

High angular–annular dark field-scanning TEM (HAADF-STEM) and energy-dispersive X-ray spectroscopy (EDS) measurements were carried out with a JEOL JEM-2200FS microscope with a Quantax 400 system and a XFlash 5060 silicon-drift detector (SDD, 60 mm^2^ active area). The samples were prepared by depositing the nanocrystal dispersion on ultrathin carbon-coated 400 mesh copper grids.

The X-ray diffraction measurements were performed on a Bruker D8 Advance system equipped with Twin-Twin technology and Shield Tube Technology for the Cu source (1.5406 Å). The maximum power of the source was 1600 W. The Standard detector was a Lynxeye Strip Detector with 192 strips working with Bragg–Brentano geometry.

The fluorescence emission spectra of the pristine and irradiated nanocrystal colloids were recorded at room temperature on a Fluoromax-P Phosphorimeter (Horiba Ltd., Kyoto, Japan), employing a 150 W Xenon continuous output ozone-free lamp. For these experiments, the dispersions were placed in quartz cuvettes.

## 3. Results and Discussion

The laser-induced morphological and structural changes of the metal halide nanocrystals have been studied by irradiating their colloidal nanocrystal solution with a high repetition rate Yb:KGW femtosecond laser system. Figure 2 presents the pristine, hexagonally shaped Cs_4_PbBr_6_/CsPbBr_3_ nanocrystals used for the irradiation experiments. The nanocrystals were synthesized with a low-temperature reprecipitation-based protocol reported earlier [21,33]. In our previous work, it had been shown that a secondary CsPbBr_3_ phase is present in the form of inclusions in the Cs_4_PbBr_6_ nanohexagons (Appendix A) [21]. The HRTEM analysis, together with the respective FFT analysis, revealed the high crystallinity of the nanohexagons, and evidenced lattice planes with a distance of 7.2 Å, which correspond to that of the (012) planes of the Cs_4_PbBr_6_ crystal structure (Appendix A). This is a unique crystallographic feature of the rhombohedral Cs_4_PbBr_6_ phase and is missing from the orthorhombic, tetragonal, and cubic polymorphs of the CsPbBr_3_ crystal structure. The slightly expanded d-spacing compared to that of the reference crystal structure (ICSD ID 025124, d = 7.0 Å) can be attributed to the growth of the secondary CsPbBr_3_ phase on top or inside the Cs_4_PbBr_6_ domain, which is confirmed also from the additional diffraction spots in the FFT pattern and coincide with the (110) crystal planes of the cubic CsPbBr_3_ crystal structure (ICSD ID 29073). The interplanar spacing is 6.0 Å in the case of the inclusions.

The pristine nanohexagons colloidal solution featured a yellowish color and a bright green emission under UV irradiation (Figure 2, upper insets) which is a strong indication of the presence of a secondary CsPbBr_3_ phase inside the non-luminescent Cs_4_PbBr_6_ one. A single peak was observed and centered at 515 nm (Figure 2, bottom inset), while the full-width half maximum (FWHM) (23 nm) of this peak indicated a narrow size distribution of the CsPbBr_3_ in the Cs_4_PbBr_6_ nanohexagons. The FWHM value of the included phase is comparable with that of CsPbBr_3_ single-phase nanocrystals synthesized with the room temperature precipitation method [34].

The nanohexagon colloidal solution was irradiated using a 513 nm laser wavelength at two different fluences, namely a high (129 mJ/cm^2^) and a low one (0.5 mJ/cm^2^). In each fluence, the impact of the number of pulses of the irradiation on the structure/morphology, as well as the PL properties, was evaluated.

### 3.1. Irradiation at the Low Laser Fluence

The nanocrystal solution remained PL active, upon the irradiation with lower than 10^6^ pulses at the low fluence of 0.5 mJ/cm^2^, without obvious differences in either its coloration or PL intensity. The respective PL spectra evidenced that the PL peak of the irradiated colloids was not shifted at all upon such irradiation conditions, while the intensity was comparable to that of the pristine ones (Figure 3a). On the other hand, the PL intensity was significantly decreased upon irradiation for longer periods (1.8 to 57.6 × 10^6^ pulses) (Figure 3b). Moreover, the FWHM of the PL peak progressively increased, whereas at highest irradiation doses, the main peak was separated in two, indicating the morphological or composition alteration of the perovskite nanocrystals. To decipher the underlying mechanism, the morphological evolution of the metal halide nanocrystals upon irradiation was carefully examined by TEM (Figure 3c–h and Appendix A). Low magnification images revealed that the nanocrystals irradiated with a small number of pulses seemed to be morphologically unaffected, while at a moderate number of pulses the nanocrystals started to aggregate. At the highest irradiation doses used, when two separated PL peaks were observed, such agglomeration became more pronounced, the morphology of the nanohexagons was totally altered and small sections appeared to have been detached. The photo-induced anion exchange of the metal-halide nanocrystals dispersed in a Cl-containing solvent has been reported by Parobec et al. [30]. In this process, the anion exchange was originated from the in-situ production of halide anions through the reductive dissociation of the solvent molecules following the interfacial electron transfer from the photoexcited CsPbX_3_ nanocrystals.

### 3.2. Irradiation at the High Laser Fluence

The laser fluence of 129 mJ/cm^2^ has been selected among the three values tested (92, 129, and 165 mJ/cm^2^), according to the quality of the crystals from the TEM images after the irradiation with 21.6 ×10^6^ pulses. The optimum nanoplatelets morphology has been observed for the laser fluence of 129 mJ/cm^2^, which resulted in well-formed nanoplatelets with sharp edges and high crystallinity. At smaller (92 mJ/cm^2^) and larger (165 mJ/cm^2^) fluence, small sections removed from the nanocrystals or completely destroyed particles have been observed, respectively (Figure 4). The selection process is described in the Appendix A.

The PL properties of the nanohexagons were significantly changed upon the irradiation with a laser fluence of 129 mJ/cm^2^, indicating extended morphological or compositional modifications (Figure 5a and Appendix A). In particular, the PL peak was separated in two peaks for 1.8 × 10^6^ pulses, in contrast with what we observed for the lower fluence tested, where only a single peak was observed for the same irradiation time (Figure 3b, dark cyan curve). The TEM images of the dispersions, following the irradiation with 1.8 × 10^6^ pulses, indicated that the appearance of the two peaks in the PL spectrum (Figure 5b) could be attributed to the exfoliation of the primary nanohexagons to thinner ones, as is evident from Appendix A. Subsequently to the exfoliation of the primary nanohexagons, which took place from 1.8 to 3.6 × 10^6^ pulses (Figure 5b,c), the particles started to fragment into smaller cubic particles following irradiation with 14.4 × 10^6^ pulses (Figure 5d,e). Finally, such small cubes were enlarged to nanoplatelet-type morphologies and subsequently to nanosheets for an irradiation time longer than 14.4 × 10^6^ pulses (Figure 5f,g). Even microsheets were observed for much longer irradiation times, but the dimensions and the thickness of these structures were not completely reproducible from one experiment to the other. Some examples of such microstructures are presented in Appendix A. The calculated FFT patterns obtained from the microsheets indicated their good crystallinity (Appendix A, left figure), while the respective PL peak (412 nm) (Appendix A, right inset) was blue-shifted compared to that of nanoplatelets (420 nm) (Figure 5).

Moreover, the EDS spectroscopy revealed that, in addition to morphological changes, a partial anion exchange takes place almost simultaneously, due to their dispersion in chlorinated solvent for the irradiation experiments (Figure 6a). In particular, upon irradiation with 1.8 × 10^6^ pulses (30 s), 21.7% of atomic percentage corresponds to chlorine at the region of interest (ROI) (Figure 6a). When the irradiation time was less than 0.6 × 10^6^ pulses (10 s), the atomic percentage of chlorine was limited to 2.6% (Appendix A). The anion exchange was also confirmed by the HRTEM analysis of an individual nanohexagon (Figure 6b). The d-spacing of the (012) crystal planes was reduced compared to the primary nanohexagons from 7.2 to 7.0 Å. This reduction of the lattice spacing is consistent with previous results on anion-exchanged CsPbBr_3_ nanocrystals. The XRD diffraction peaks were shifted to higher angles when Br^−^ is exchanged with Cl^−^ [35].

It is important to note that the irradiation duration required for the formation of the morphologies observed here is only a few minutes. In particular, the nanohexagons exfoliation occurred within 30 s (1.8 × 10^6^ pulses, Figure 4b and Appendix A), while the fragmentation was initiated in 2 min (7.2 × 10^6^ pulses, Figure 2d and Appendix A) and resulted in a dispersion of only cubic-shaped nanocrystals in 4 min (14.4 × 10^6^ pulses, Figure 2e and Appendix A). Finally, the platelet-like morphologies started to be formed in 6 min (21.6 × 10^6^ pulses, Figure 5f). In addition, as indicated from EDS, the anion exchange was very fast and occurred from the first 10 s of the irradiation (Appendix A). The above irradiation times are much shorter than in previously reported photo-induced processes in similar materials [29,30].

Despite the morphological transformation, structural changes have also been observed through the analysis of the FFT patterns obtained from the HRTEM images. As already mentioned, partial anion exchange was taking place in the nanohexagons even in the early irradiation times and the lattice spacing of the (012) Cs_4_PbBr_6_ crystal planes was reduced from 7.2 Å in the initial nanocrystals to 6.9 Å for 1.8 × 10^6^ pulses for most of the particles (Figure 7a), as the Cl^−^ atoms were increased (Cs_4_PbBr_6−x_Cl_x_). Furthermore, the analysis of the FFT patterns of small nanocubes obtained by the fragmentation occurred at 7.2 × 10^6^ pulses showed a CsPbBr_3_ cubic phase with reduced d-spacing values due to the partial anion exchange (CsPbBr_3−x_Cl_x_) (Figure 7b). The d-spacings in this case are 5.8 and 4.1 Å for the planes (100) and (110) respectively, for most of the nanocubes, while small variations on these values can be originated for different Cl^−^ content. Any nanohexagon that retained its morphology at that irradiation time also kept the d-spacing equal to 7 Å (Appendix A), similar to the nanohexagons irradiated for a shorter time. Τhe first indication of full anion exchange was at 14.4 × 10^6^ pulses and the number of these particles progressively increased with the irradiation time. Most of the CsPbCl_3_ (ID 201250) particles were observed after 21.6 × 10^6^ pulses. Some of the particles retained the CsPbBr_3−x_Cl_x_ or even the CsPbBr_3_ compositions but the number of these particles was reduced with the time of the irradiation, and this is also obvious from the PL spectra, where the initial peak became less intense but never faded completely. In terms of compositions associated to the different morphologies, we can state that the nanohexagons were Cs_4_PbBr_6−x_Cl_x_, the small nanocubes were CsPbBr_3−x_Cl_x_, the larger nanoplatelets were CsPbCl_3_, while the nanosheet-like particles were essentially CsPbBr_3_ (Figure 7 and Appendix A).

### 3.3. Laser-Triggered Proposed Transformation Mechanism

According to the results presented here, the laser wavelength, photon energy and the number of pulses are among the parameters that can affect the optical and thermal properties of the metal halide nanohexagons and determine the physical phenomena taking place during the irradiation process. The laser-triggered transformations due to the efficient optical absorption of the nanohexagons under the 513 nm wavelength laser irradiation of 129 mJ/cm^2^ fluence can be summarized as follows: (i) exfoliation in thinner nanohexagons and partial anion exchange, (ii) fragmentation in smaller nanocubes and partial anion exchange, (iii) side-by-side-oriented attachment, fusion, and formation of the nanoplatelets, partial and in some particles complete anion exchange, and (iv) side-by-side attachment, fusion, and formation of nanosheet-like morphologies (Figure 7). Additional experiments confirmed that IR wavelengths and 129 mJ/cm^2^ fluence, similar to the previous experiments, are not sufficient to alter their morphology, possibly due to the poor optical absorption of the nanocrystals at these wavelengths. The PL enhancement in this case can be attributed to the enlargement of the CsPbBr_3_ inclusions in the nanohexagons.

The nanohexagons, due to the precipitation method used for their synthesis and the long period required for them to be formed (7 days), seem to arrange in stacking-like structures and 30 s are sufficient for the laser to exfoliate them. Laser-irradiation has also been reported from different research group as an efficient method to exfoliate layered-structured materials [4,36,37] in liquid environments. Following the thinning of the nanohexagons as the time of the irradiation increases, sufficient energy is absorbed and induces stress to the structure of the nanohexagons, and their fragmentation is initiated. In two minutes, nanohexagons together with small nanocubes are found in the dispersion. The coexistence of the initial nanohexagons together with the fragmented ones in the early time of the irradiation was also reported by Schaumberg et al., as the laser beam, due to its gaussian shape, does not react with the same manner in the whole dispersion [2]. The fragmented nanocrystals had a broad size distribution, which is also observed in the latter report [2]. Furthermore, by selectively tuning the irradiation conditions, only a few minutes were enough time to have the fragmentation of the 100 nm nanohexagons, in contrast to previous reports in which the fragmentation process could take up to three hours [29]. Then, the fragmented species from the Cs_4_PbBr_6_/CsPbBr_3_ were recrystallized again and partially chlorinated to form the CsPbBr_3−x_Cl_x_ nanocubes, in contrast to the fragmentation of metal halides reported by Dong et al. and Amendola et al., where the laser did not induce any structural transformation [29,38].

Following the fragmentation, the nanocubes were fused together due to the laser-induced destabilization of the surface ligands to form nanoplatelets or nanosheets, if the sample was irradiated for longer times. Furthermore, the photo-induced anion exchange was faster than in the report by Parobek et al., which studied small cubes (with sizes around 15–20 nm) irradiated with a 405 nm wavelength laser [30]. In that report, the PL peak was shifted to 420 nm after 19 min, while in our work this took 12 min.

This laser-induced nanocrystals transformation method does not have the high yield in terms of material production compared to the precipitation or the hot-injection methods, nor the best homogeneity; however, it is rapid and clean, without the need to add a second chemical compound or an extra purification step and can be scaled up after the design of the irradiation process. In such a process, the final nanocrystal stoichiometry could be tuned by selecting only the proper dispersion solvent (dichlorobenzene, diiodobenzene or dibromobenzene) in which the irradiation is taking place.

## 4. Conclusions

In summary, we have demonstrated a rapid photo-induced approach to tune the structural and, consequently, the optical properties of metal halide nanocrystals in solution by irradiation with a femtosecond laser for a few minutes. The laser fluence and wavelength dependence of the final morphology and structure of the nanocrystals have been investigated. An exfoliation of the initial Cs_4_PbBr_6_/CsPbBr_3_ nanohexagons took place in the first 30 s, then these particles started to fragment into smaller cubic particles until 4 min, and finally, these small cubes were enlarged to nanoplatelet-type morphologies until 12 min. Sheets of lateral sizes of around 1–1.5 μm of good crystallinity could also be obtained for longer irradiation times. A small amount of Br^−^ was exchanged in the structure, with Cl^−^ originating from the chlorinated solvent in the first minutes (CsPbBr_3−x_Cl_x_), while a complete anion exchange occurred in most of the nanocrystals after 4 min (CsPbCl_3_). The PL peak was observed to blue-shift from 515 of the initial nanohexagons to 412 nm. The quality of these structures could be modified by tuning the laser fluence. Well-formed particles with sharp edges have been obtained with laser fluence of 129 mJ/cm^2^, while partially etched or melted-like structures have been obtained using smaller or larger fluences, respectively. The use of chlorinated solvent as dispersive medium also gives the opportunity to study photo-induced anion exchange mechanisms in these structures.

## Figures and Tables

**Figure 1 nanomaterials-12-00703-f001:**
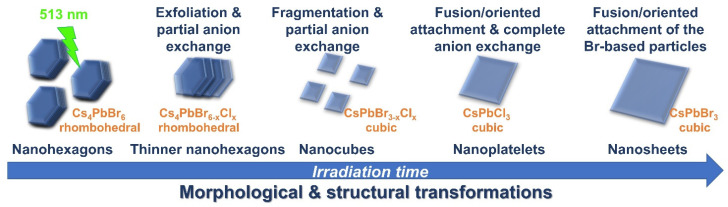
Photo-triggered morphological and structural transformations of metal halide nanohexagons.

**Figure 2 nanomaterials-12-00703-f002:**
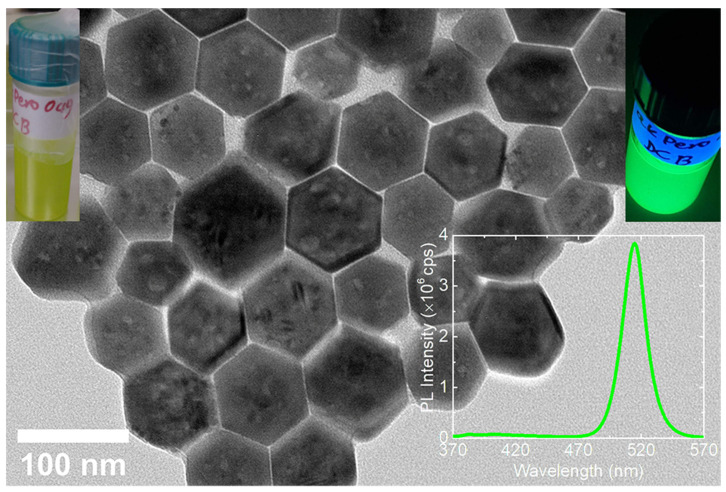
Low magnification transmission electron microscopy (TEM) image of the pristine cesium lead bromide nanohexagons before the laser irradiation. Insets: Photo of DCB-based solution of the nanohexagons (**upper left**), the same solution under UV lamp (**upper right**) and its PL spectrum (**bottom right**).

**Figure 3 nanomaterials-12-00703-f003:**
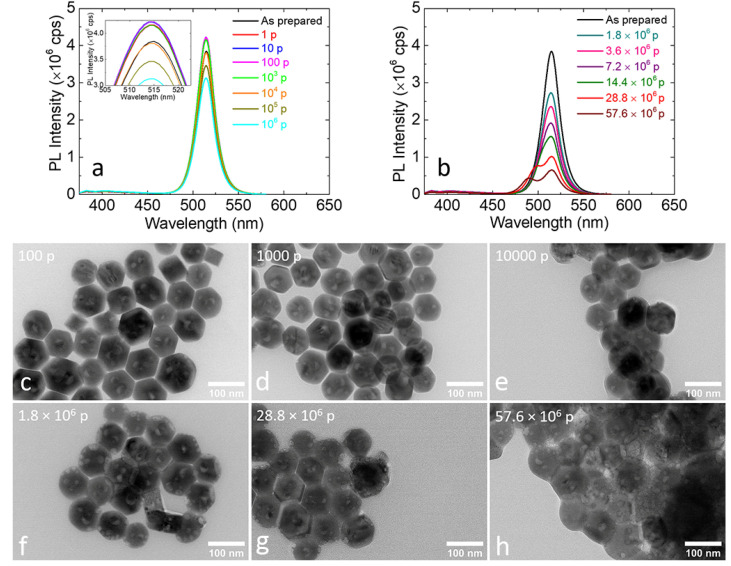
Photoluminescence spectra of the metal halide perovskite nanohexagons dispersed in DCB and irradiated with a laser fluence of 0.5 mJ/cm^2^ and number of pulses from single to 10^6^ (**a**) and from (1.8 to 57.6) × 10^6^ pulses (**b**). Representative low-magnification TEM images of the irradiated nanocrystals for different number of pulses (**c**–**h**). The irradiation was carried out with a femtosecond laser of 513 nm wavelength.

**Figure 4 nanomaterials-12-00703-f004:**
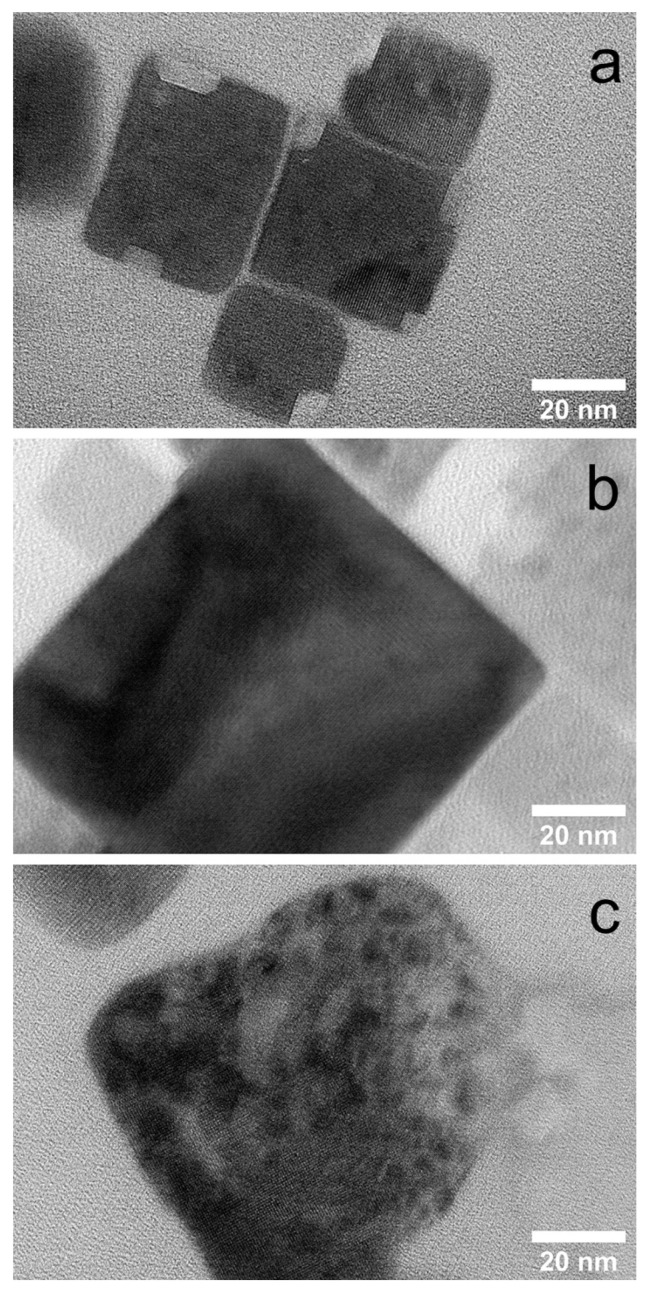
HRTEM images of the DCB-based nanohexagon solutions irradiated with 92 (**a**), 129 (**b**) and 165 (**c**) mJ/cm^2^ fluence and 21.6 × 10^6^ pulses.

**Figure 5 nanomaterials-12-00703-f005:**
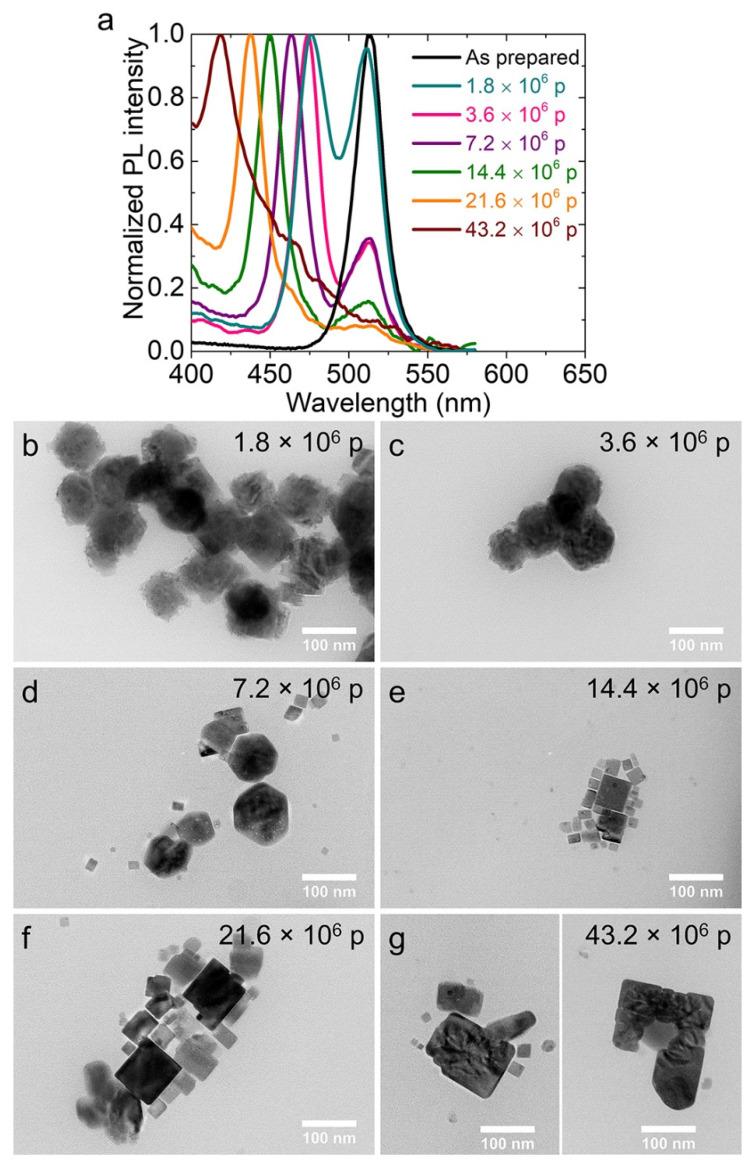
Photoluminescence spectra of the DCB-based nanohexagon solution irradiated with 129 mJ/cm^2^ for 1.8 to 43.6 × 10^6^ number of pulses (**a**). Low magnification TEM images of the irradiated nanocrystals for the same irradiation durations (**b**–**g**). The Figure 5g are two particles from two different TEM grid regions. The irradiation was carried out with a fs laser of 513 nm wavelength.

**Figure 6 nanomaterials-12-00703-f006:**
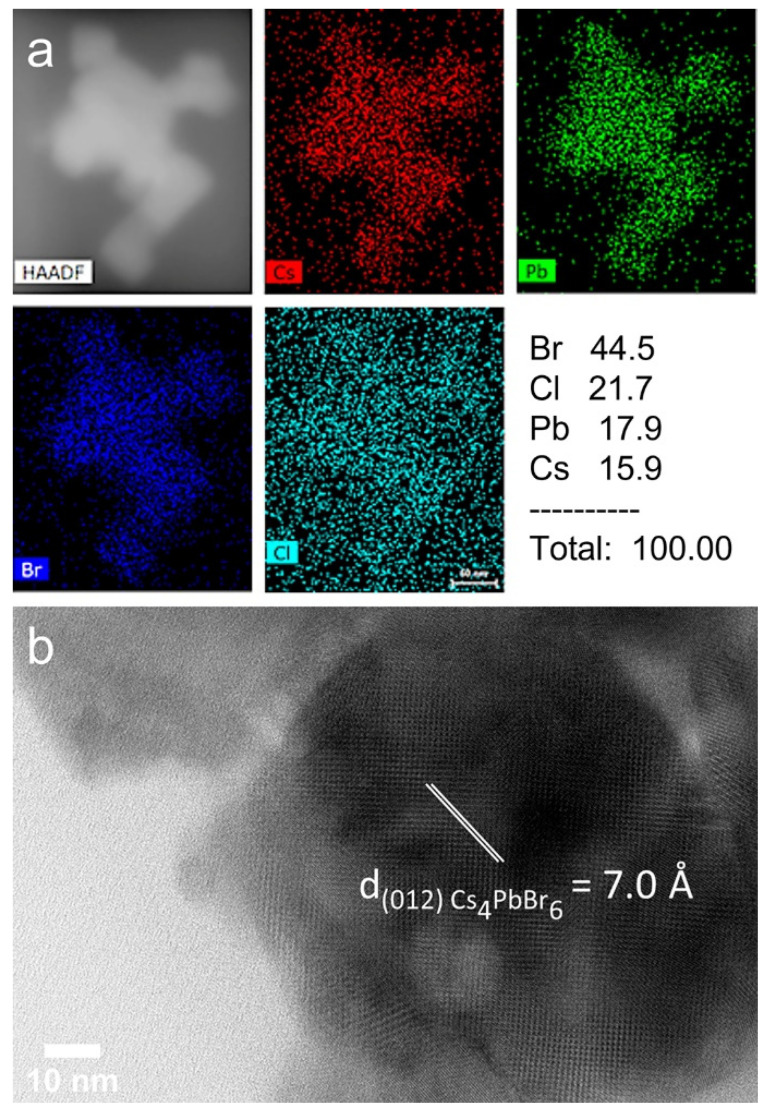
EDS mapping (**a**) and HRTEM images (**b**) of the nanohexagon DCB-based solution following irradiation with 1.8 × 10^6^ pulses.

**Figure 7 nanomaterials-12-00703-f007:**
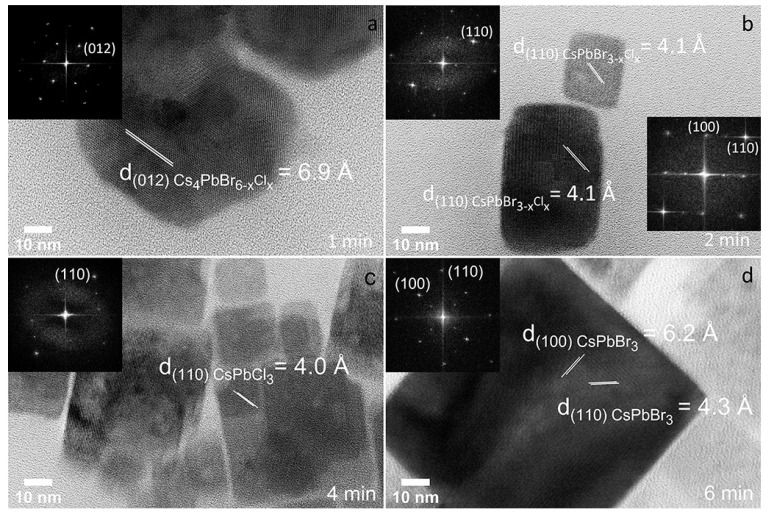
HRTEM images of the Cs_4_PbBr_6−x_Cl_x_ nanohexagons after 3.6 × 10^6^ pulses (**a**), CsPbBr_3−x_Cl_x_ nanocubes after 7.2 × 10^6^ pulses (**b**), CsPbCl_3_ nanoplatelets after 14.4 × 10^6^ pulses (**c**) and Cs_4_PbBr_3_ nanosheets after 21.6 × 10^6^ pulses (**d**).

## Data Availability

Not applicable.

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
