# Peer review of "Laser-Induced Morphological and Structural Changes of Cesium Lead Bromide Nanocrystals"

_nanomaterials, 2022, doi:10.3390/nano12040703_

Round 1
Reviewer 1 Report
In this work, the authors investigate the effect of laser irradiation on CsPbBr nanocrystals. They find that upon irradiation with 515 nm wavelength pulses, on a scale form seconds to minutes, the hexagonal crystals first undergo exfoliation and fragmentation, before than recombining into larger nanoplatelets upon longer exposure. Since the crystals are suspended in dichlorobenzene solvent, anion exchange also occurs resulting in the formation of a chlorinated perovskite phase as well. These findings are supported by extensive PL and TEM measurements. Even though the degree of deliberate control over the structural changes in the crystals is not yet fully developed, the paper is overall well presented and interesting, so I recommend it for publication after a few smaller points are addressed.
- What are the statistics of the experiment performed? On how many samples as laser irradiation attempted? This should be noted in the materials and methods section.
- PL measurements as a function of irradiation time are only shown normaliz N unnormalized plot should also be provided to ae how much recrystalization into platelets affects the PL yield.
- Can the PL blue shift be explained by formation of a chlorinated phase upon ion exchange?
- Are there changes in the crystals absorbance as a result of irradiation?
- What is the irradiation time of figure 5g?
- Please check the supplemental information text for grammar
Reviewer 2 Report
In this work the authors study morphological, structural and Br-Cl anion exchange by fs-laser ablation of Cs4PbBr6/CsPbBr3 nanohexagons dispersed in dichlorobenzene. The structural characterization of transformed nanocrystals under different synthesis conditions (laser fluence and ablation time under 513nm irradiation) demonstrate the main conclusions claimed by the authors (different stages of the ablation process) and consistent with emission spectra (reflecting the Br-Cl anion exchange). Moreover, the paper is well organized and well written, hence deserves publication.
Only some minor remarks:
- Indicate the conditions of the ablation process. The authors refer to a previous paper, but it is better to indicate also in this work how the ablation process was carried out. Particularly, if the whole solution volume is swept by laser using the mirrors or the volume (or if the volume is in the order of the laser spot size) …
- The authors claim dimensionality change, but this is more difficult to be demonstrate and it is necessary to have dimensions smaller than exciton size as quantum dots or nanosheets of small thickness (in these cases absorption + photoluminescence spectra are necessary). May be better to claim morphology and Br-Cl anion exchange that are clearly demonstrated.
Author Response
Response to Reviewer 2 Comments
In this work the authors study morphological, structural and Br-Cl anion exchange by fs-laser ablation of Cs4PbBr6/CsPbBr3 nanohexagons dispersed in dichlorobenzene. The structural characterization of transformed nanocrystals under different synthesis conditions (laser fluence and ablation time under 513 nm irradiation) demonstrate the main conclusions claimed by the authors (different stages of the ablation process) and consistent with emission spectra (reflecting the Br-Cl anion exchange). Moreover, the paper is well organized and well written, hence deserves publication.
Only some minor remarks:
Point 1. Indicate the conditions of the ablation process. The authors refer to a previous paper, but it is better to indicate also in this work how the ablation process was carried out. Particularly, if the whole solution volume is swept by laser using the mirrors or the volume (or if the volume is in the order of the laser spot size) …
Response 1: In order to be easier for the reader to find the exactly experimental conditions we added in the “Material and Methods section” in the subsection “ii) Laser irradiation experiment: Set up and irradiation conditions” the description of the set up. The volume is in the order of the laser spot size.
Point 2. The authors claim dimensionality change, but this is more difficult to be demonstrate and it is necessary to have dimensions smaller than exciton size as quantum dots or nanosheets of small thickness (in these cases absorption + photoluminescence spectra are necessary). May be better to claim morphology and Br-Cl anion exchange that are clearly demonstrated.
Response 2: In order to avoid misconceptions according to the reviewer comment, we replaced the “dimensionality” with the term “structural” or “structure” in the whole text. In particular we changed the following:
- Page 2, line 81: …of the impact on structure and morphology
- Page 3, line 89: Fig. 1 caption, Photo-triggered morphological and structural transformations
- Page 9, line 268: Despite the morphological transformation, structural changes…
- Page 11, line 346: …of the final morphology and structure of the nanocrystals.
Reviewer 3 Report
The authors of this paper have employed a 513 nm fs laser to induce different transformations in large nanocrystals, in which two phases coexist in the same particle (Cs4PbBr6/CsPbBr3 nanohexagons of 90-100 nm), dispersed in dichlorobenzene. And summed up the transformation law through the research, these transformations include: i) exfoliation of the primary nanohexagons and partial anion exchange; ii) fragmentation in smaller nanocubes and partial anion exchange; iii) side-by-side-oriented attachment, fusion and formation of nanoplatelets and complete anion exchange; iv) side-by-side attachment, fusion and formation of nanosheets.
The paper is rich in content with sufficient data to illustrate the experimental conclusions. It has certain theoretical significance and experimental application value, and the conclusion is clear and its analysis is reasonable. The results are novel enough to be published. I recommend the publication of this manuscript after minor revision. Here are a few recommendations for the authorsto improve this manuscript. This work would be greatly improved if the authors could discuss in depth on the internal mechanism of crystal phase transformation. Figure 1 and Figure 3(a) are not clear enough, it is highly recommended to modify both of them.
Author Response
Response to Reviewer 3 Comments
The authors of this paper have employed a 513 nm fs laser to induce different transformations in large nanocrystals, in which two phases coexist in the same particle (Cs4PbBr6/CsPbBr3 nanohexagons of 90-100 nm), dispersed in dichlorobenzene. And summed up the transformation law through the research, these transformations include: i) exfoliation of the primary nanohexagons and partial anion exchange; ii) fragmentation in smaller nanocubes and partial anion exchange; iii) side-by-side-oriented attachment, fusion and formation of nanoplatelets and complete anion exchange; iv) side-by-side attachment, fusion and formation of nanosheets.
The paper is rich in content with sufficient data to illustrate the experimental conclusions. It has certain theoretical significance and experimental application value, and the conclusion is clear and its analysis is reasonable. The results are novel enough to be published. I recommend the publication of this manuscript after minor revision. Here are a few recommendations for the authors to improve this manuscript.
Point 1. This work would be greatly improved if the authors could discuss in depth on the internal mechanism of crystal phase transformation. Figure 1 and Figure 3(a) are not clear enough, it is highly recommended to modify both of them.
Response 1: The internal mechanism of the crystal phase transformation is described in the section 3.3 and it is depicted in Figure 1 presenting the different transformation steps as supported by the experimental data. A more in-depth discussion could require more experimental data on the crystal structures and theoretical calculations which is planned for the future.
The curves in the Figure 3a are overlapping for this we include the magnified one as inset.
Reviewer 4 Report
Kostopoulou and co-workers studied the morphology and structure transformation of Cs4PbBr6/CsPbBr3 nanohexagons under laser irradiation. More characterizations and experiments were performed for analyzing the structure-photoluminescence relationship and proposing the laser-triggered transformation mechanism. It is an interesting topic, and can be accepted after minor revision.
1. As the results shown in this work, the PL peak of NCs was split in two peaks after irradiation under a laser fluence of 129 mJ/cm2 for 1.8 × 106 pulses. What is the main reason to that phenomenon? Is it the exfoliation of primary nanohexagons or the emergence of CsPbBr3-xClx NCs? Maybe a controlled experiment is needed by changing the solvent that would not lead to anion exchange.
2. The authors observed structure change of Cs4PbBr6/CsPbBr3 NCs under 129 mJ/cm2 laser irradiation by analyzing FFT patterns, however, it would be more convinced if the whole process is also monitored by XRD.
3. Figure 3c consists of 6 TEM images. This reviewer suggests the authors use different letters to label them other than only one.
Author Response
Response to Reviewer 4 Comments
Kostopoulou and co-workers studied the morphology and structure transformation of Cs4PbBr6/CsPbBr3 nanohexagons under laser irradiation. More characterizations and experiments were performed for analyzing the structure-photoluminescence relationship and proposing the laser-triggered transformation mechanism. It is an interesting topic, and can be accepted after minor revision.
Point 1. As the results shown in this work, the PL peak of NCs was split in two peaks after irradiation under a laser fluence of 129 mJ/cm2 for 1.8 × 106 pulses. What is the main reason to that phenomenon? Is it the exfoliation of primary nanohexagons or the emergence of CsPbBr3-xClx NCs? Maybe a controlled experiment is needed by changing the solvent that would not lead to anion exchange.
Response 1: Both mechanisms (exfoliation and anion exchange) contribute to the fast splitting of the PL peak. We did also the irradiation in toluene instead of dichlorobenzene as control experiment and only exfoliation is observed and splitting of the PL peak, but this peak does not shift for long irradiation durations. Also, carbonization of toluene was observed inducing problems to the TEM observation.
Point 2. The authors observed structure change of Cs4PbBr6/CsPbBr3 NCs under 129 mJ/cm2 laser irradiation by analyzing FFT patterns, however, it would be more convinced if the whole process is also monitored by XRD.
Response 2: This is correct that it could be more convincing if the phase transformations were supported also by XRD data, but this was not feasible due to technical restrictions. In order to obtain the best homogeneity, we diluted the dispersions a lot and the whole volume was 500 μl. This quantity was really small for a XRD spectrum of good quality , and we didn’t want to mix different irradiated samples.
Point 3. Figure 3c consists of 6 TEM images. This reviewer suggests the authors use different letters to label them other than only one.
Response 3: Thank you for this suggestion. We added different letters in each figure.